# The Mechanisms Underlying Salt Resistance Mediated by Exogenous Application of 24-Epibrassinolide in Peanut

**DOI:** 10.3390/ijms23126376

**Published:** 2022-06-07

**Authors:** Wenjiao Li, Jie Sun, Xiaoqian Zhang, Naveed Ahmad, Lei Hou, Chuanzhi Zhao, Jiaowen Pan, Ruizheng Tian, Xingjun Wang, Shuzhen Zhao

**Affiliations:** 1Institute of Crop Germplasm Resources, Shandong Academy of Agricultural Sciences, Shandong Provincial Key Laboratory of Crop Genetic Improvement, Ecology and Physiology, Jinan 250100, China; 18790626683@163.com (W.L.); sunjie9811@163.com (J.S.); zhang033097@163.com (X.Z.); naveedjlau@gmail.com (N.A.); houlei9042@163.com (L.H.); chuanzhiz@126.com (C.Z.); jwpan01@126.com (J.P.); aarongood@163.com (R.T.); xingjunw@hotmail.com (X.W.); 2College of Life Sciences, Shandong Normal University, Jinan 250014, China

**Keywords:** *Arachis hypogaea*, 24-epibrassinolide, salt stress, osmotic adjustment, RNA-Seq

## Abstract

Peanut is one of the most important oil crops in the world, the growth and productivity of which are severely affected by salt stress. 24-epibrassinolide (EBL) plays an important role in stress resistances. However, the roles of exogenous EBL on the salt tolerance of peanut remain unclear. In this study, peanut seedlings treated with 150 mM NaCl and with or without EBL spray were performed to investigate the roles of EBL on salt resistance. Under 150 mM NaCl conditions, foliar application of 0.1 µM EBL increased the activity of catalase and thereby could eliminate reactive oxygen species (ROS). Similarly, EBL application promoted the accumulation of proline and soluble sugar, thus maintaining osmotic balance. Furthermore, foliar EBL spray enhanced the total chlorophyll content and high photosynthesis capacity. Transcriptome analysis showed that under NaCl stress, EBL treatment up-regulated expression levels of genes encoding peroxisomal nicotinamide adenine dinucleotide carrier (*PMP34*), probable sucrose-phosphate synthase 2 (*SPS2*) beta-fructofuranosidase (*BFRUCT1*) and Na^+^/H^+^ antiporters (*NHX7* and *NHX8)*, while down-regulated proline dehydrogenase 2 (*PRODH*). These findings provide valuable resources for salt resistance study in peanut and lay the foundation for using BR to enhance salt tolerance during peanut production.

## 1. Introduction

Soil salinity is one major abiotic stress that affects plant growth and development and further limits crop productivity. About 950 million hectares of land were affected by salinity in the world [1]. The increase of salt concentration in the plant growth environment will lead to the increase of Na^+^ and Cl^−^ in plants, and induce oxidative stress, osmotic stress and ionic toxicity [2]. Oxidative stress leads to the accumulation of reactive oxygen species (ROS), including hydrogen peroxide (H_2_O_2_), superoxide (O_2_^−^), singlet oxygen (^1^O_2_) and the hydroxyl radical (^−^OH) [3]. The non-enzymatic and enzymatic antioxidant defense systems can effectively remove ROS and alleviate oxidative stress. The antioxidant enzymes mainly include superoxide dismutase (SOD), peroxidase (POD), catalase (CAT) and Ascorbate peroxidase (APX). Under abiotic stress, these enzymes could reduce the level of ROS to a certain extent and maintain the normal growth of plants [4]. A high level of ROS causes peroxidation of membrane lipid and producing malondialdehyde (MDA) [5]. Salt stress leads to osmotic stress on plants, and eventually affected plant growth and development [6]. Some osmolytes such as proline, soluble protein, betaine, soluble sugar and polyamine could be accumulated in plant cell under stress condition, which regulated osmotic pressure, thereby reduced water potential and maintained normal plant growth [7]. Salt stress led to ion imbalance, increased in Na^+^ content and decreased K^+^, Mg^2+^ and Ca^2+^ contents in plants, resulting in Na^+^ toxicity [8,9]. The salt overly sensitive (SOS) pathway was essential for maintaining Na^+^ and K^+^ homeostasis in the cytoplasm [10]. High Na^+^ concentration triggered a calcium signal, and stimulated SOS2-SOS3 protein kinase complex, and then activated Na^+^/H^+^ antiporter SOS1 to eliminate excess Na^+^ in plant cells [11].

Brassinosteroids (BRs) are essential for plant growth and development as well as environmental adaptation [12]. The signal transduction pathway of BRs has been studied extensively in *Arabidopsis thaliana.* Brassinosteroid-insensitive 1 (BRI1) is a BR receptor and senses BR signal [13]. BR binds and activates BRI1, and then triggers the formation of a brassinosteroid insensitive 1-associate-receptor kinase 1 (BRI1-BAK1) heterodimer. Activated BRI1 is separated from BR-signal-inhibitor (BKI1) and transmits BR signal to BR-signaling-kinase 1 (BSK1) and constructive-differential-growth 1 (CDG1) through a series of phosphorylation reactions. BSK1 activates phosphatase BRI1-suppressor 1 (BSU1), phosphorylates brassinosteroid-insensitive 2 (BIN2) to culminate in its ubiquitination and degradation. Upon BIN2 inactivation, the activity and stability of plant specific transcription factors brassinazole-resistant 1 (BZR1) or BRI1-emssuppressor 1 (BES1) are promoted, which directly controls the transcription of genes downstream of BR [14]. Previous studies have shown that BR not only plays an important role in plant growth and development, but also improves plant tolerance to abiotic stress. However, the molecular mechanism in which BR signal responds to abiotic stress in peanut remain unknown.

In recent years, there has been extensive research on the response and mechanism of BR to abiotic stress. 24-epibrassinolide (EBL), one of the most active synthetic analogs of the BRs family, improved salt stress tolerance in soybean [15], canola [16], potato [17] and perennial ryegrass [18]. BR alleviated salt stress by improving the antioxidant activity and proline content [19,20]. Under stress, foliar spray with BL significantly improved the photosynthetic attributes, nutrient partitioning, alleviated ion toxicity and oxidative damage [21]. BRI1, BIN2 and BZR1 are the key factors of BR signaling pathway. Arabidopsis mutant *bri 1-9* was more sensitive to salt in comparison to wild-type seedlings [22]. BZR1 positively regulated the salt tolerance of tomato by up-regulating the expression of multiple stress-related genes [23]. BIN2 negatively regulated salt stress tolerance through inhibiting the function of SOS2 by phosphorylation [24]. However, the response mechanism of BR involved in salt stress has not been fully analyzed, and it is not clear which factor plays a decisive role. In the present study, we explored the physiological and molecular mechanisms of EBL application mediated salt stress tolerance in peanut. Our results provide valuable information for improving management strategy when peanut grown in saline soils.

## 2. Results

### 2.1. Effects of EBL Treatments on the Growth of Peanut Seedling

Peanut seedlings of 12 d were treated with five concentrations of NaCl including 0 mM, 50 mM, 100 mM, 150 mM and 200 mM. Based on the results including plant height and MDA content under different NaCl concentrations, 150 mM NaCl was selected as the stress concentration (Appendix A). Under 150 mM NaCl condition, we also carried out treatments with five EBL concentration, 0 µM, 0.001 µM, 0.01 µM, 0.1 µM and 1 µM EBL, respectively. According to the results, 0.1 µM EBL was selected for further experiments (Appendix A). Peanut seedlings of 12 d were divided into four groups for treatments, CK (normal conditions), N (treated with 150 mM NaCl), NE (treated with 150 mM NaCl and 0.1 µM EBL) and E (treated with 0.1 µM EBL). Results showed that the main stem height, root length, fresh weight and dry weight of peanut seedlings were significantly inhibited under 150 mM NaCl treatment comparing to the control. Compared with 150 mM NaCl treatment alone, the main stem height, fresh weight and dry weight of peanut seedling were improved markedly after simultaneous treatment with 0.1 µM EBL. However, EBL applications did not show any effect on NaCl-induced root growth inhibition (Figure 1).

### 2.2. Effects of EBL Treatments on Production of O_2_^−^ and H_2_O_2_ of Peanut Seedling

Under abiotic stresses, over accumulation of ROS including superoxide anion radical (O_2_^−^) and hydrogen peroxide (H_2_O_2_), which could cause plant cellular oxidative damage. We used the histochemical staining of nitroblue tetrazolium (NBT) to detect O_2_^−^ level under salinity stress. As shown in Figure 2A, the blue color intensity in leaves treated with 150 mM NaCl was much higher than those of control plants, indicating more O_2_^−^ in NaCl treated plants than that in control plants. However, no obvious difference was detected in NE and control plants (Figure 2A). After NaCl treatment, the H_2_O_2_ content was higher than the control. Exogenous EBL application significantly reduced the H_2_O_2_ content caused by NaCl (Figure 2B). These results indicated that EBL reduced ROS production under salt stress.

### 2.3. Effects of EBL Treatments on MDA Content and Antioxidant Enzyme Activities

Under salt stress, the MDA content was higher than the control, while the contents of MDA between control and NE plants were not significantly different (Figure 3A). There was no significant difference detected in SOD activity among different treatments. However, compared with the control, POD activity of seedlings under N and NE treatments increased remarkably. While POD activity under NE treatment was declined significantly compared with NaCl treatment. On the contrary, compared with the control, CAT activity was deceased with NaCl treatment. When exogenous EBL was applied under salt stress, the activity of CAT was increased compared with NaCl treatment.

### 2.4. Effects of EBL Treatments on Osmolytes Accumulation and Leaf Relative Water Content (RWC)

Osmolytes accumulation plays important roles in plant responses to abiotic stresses. Our results showed that the contents of proline and soluble sugar were significantly increased after NaCl treatment, while the soluble protein content was similar between the control and the NaCl treated plants (Figure 4). EBL treatment significantly increased the contents of proline and soluble sugar to compare with NaCl treatment and the control. A slight increase of RWC was detected in the seedlings under NE treatment, although RWC of plants under N, NE and E treatments were all reduced compared with the control (Figure 4D).

### 2.5. Effects of EBL Treatments on Chlorophyll and Carotenoids Content

NaCl treatment resulted in a slight decline in chlorophyll a and carotenoids compared with the control, while NE treatment led to a significant increase of the level of chlorophyll a and carotenoids comparing to NaCl treatment. In contrast, no significant differences were observed in the *chlorophyll* b content among the four treatments (Figure 5).

### 2.6. Effects of EBL Treatments on Chlorophyll Fluorescence

Under 150 mM NaCl treatment, the maximum photochemical quantum yield of PSII (Fv/Fm), PSII (ΦPSII), electron transfer efficiency (ETR), and photochemical quenching coefficient (qP) were decreased significantly compared with the control. While salt treatment led to an increase of non-photochemical quenching (NPQ) remarkably. EBL treatment increased the values of ΦPSII, ETR and qP, respectively, compared with only NaCl treatment. However, EBL application on plants treated by NaCl led to a decrease of NPQ (Figure 6).

### 2.7. Effects of EBL Treatments on Key Gene Expression

We found that EBL treatments modulated the expression of key genes involved in salt tolerance and BR signal pathway in peanut seedlings under salt stress (Figure 7). qPCR results showed that the expression of key genes changed the most significantly after 12 h-treatments. Under salt stress, the expression levels of *AhSOS1*, *AhDWF4* and *AhBES1* were induced and reached to its maximum after 12 h of NaCl treatment, while the expression level of *AhNHX1* was peaked after 4 h of NaCl treatment. Under salt stress, when exogenous EBL was supplemented, the expression levels of *AhNHX1*, *AhDWF4* and *AhBES1* were increased, whereas *AhSOS1* expression was decreased compare with NaCl treated plants without EBL. Therefore, peanut seedlings with 12 h treatment were selected as materials for RNA-seq sequencing to further explore the molecular mechanism of EBL alleviating salt stress of peanut.

### 2.8. Transcriptome Sequencing and Analysis of Differentially Expressed Genes (DEGs)

After peanut seedlings were treated for 12 h, RNA extracted from the leaves of CK, N, NE and E treatments were sequenced. The samples were marked as CK12_1, CK12_2, CK12_3, N12_1, N12_2, N12_3, NE12_1, NE12_2, NE12_3, E12_1, E12_2 and E12_3 (Table 1). Each sample produced an average of 1.19 Gb data, the percentages of reads with quality scores of Q20 and Q30 were higher than 98.28% and 94.72%, respectively.

### 2.9. Gene Ontology (GO) and KEGG Pathway Analysis of DEGs of N12-vs.-NE12

We performed GO enrichment analysis of N12-vs.-NE12 to explore the functions of the DEGs responded to EBL treatment (Figure 8). The identified DEGs were divided into three functional categories, including molecular function, cellular component and biological process. In the biological process term, ‘cellular process’ was the most enriched, followed by ‘metabolic process’ and ‘stimulus response’. In the cellular component term, the most significantly enriched term was ‘cells, followed by ‘cell part’, ‘organelles and membranes’. In the molecular function term, the most significantly enriched terms were ‘binding’, followed by ‘catalytic activity’, ‘transport activity’ and ‘transcription regulatory activity’.

Among 110 KEGG pathways of N12-vs.-NE12, 15 pathways were significantly enriched, with corrected *p* value ≤ 0.05 (Figure 9). DEGs were significantly enriched in photosynthetic antenna protein, alanine, aspartate and glutamate metabolism, and protein processing in endoplasmic reticulum. In addition, it was also enriched in ‘plant hormone signal transduction’, ‘glycolysis/gluconeogenesis’ and ‘starch and glucose metabolism’.

### 2.10. EBL Regulates Salt Tolerance Genes in Peanut Seedlings Exposed to Salt Stress

DEGs were selected using the standard of both |log2FC| ≥ 1 and Q value ≤ 0.05. Compared to N12 treatment, genes in NE12 treatment contained 306 up-regulated genes and 148 down-regulated genes. The Na^+^/H^+^ antiporters (NHXs) are very important players in plant salt tolerance. The expressions of *AhNHX7* and *AhNHX8* were up-regulated by EBL in peanut seedlings exposed to salt stress (Table 2). Compared with CK12, the expressions of *AhNHX7* and *AhNHX8* were down-regulated in N12; Compared with N12, the expressions of *AhNHX7* and *AhNHX8* in NE12 were up-regulated.

The RNA-seq data showed that some ROS scavenging related genes were up-regulated (Table 2). Compared with N12, the expression of peroxisome related genes was up-regulated in NE12, including *PMP34* (peroxisomal nicotinamide adenine dinucleotide carrier), *AAE11* (butyrate-CoA ligase AAE11) and *ECH2* (enoyl-CoA hydratase 2). Compared with N12, the expression of peroxidase Q coding genes was up-regulated in NE12. Furthermore, the expression of genes encoding chloroplast L-ascorbic acid peroxidase T and glutaredoxin-C6 were also up-regulated by EBL under salt stress.

Salinity imposes osmotic stress on plants by impairing water uptake and stomatal opening. Under salt stress, EBL treatment up-regulated the expression of proline related genes *PERK6* (putative proline-rich receptor-like protein kinase PERK6), *PERK9* and down-regulated *PRODH* (proline dehydrogenase 2). Compared with N12, expressions of sugar carrier protein C-like, *SPS2* (sucrose-phosphate synthase 2) and *BFRUCT1* (beta-fructofuranosidase) were up-regulated in NE12, while expression of alpha-amylase-like gene was down-regulated in NE12 (Table 2).

Phytohormones play important roles in plant growth, development, and adaptation to adverse environment including salinity. EBL treatment under salt stress up regulated the expression of cytochrome *P450 90A1* (*CYP90A1*/*CPD*), EBL receptor (*BRI1*), BR signal kinase (*BSK5*) and BR effector (*TCH4*) genes (Table 2), thus promoted BR synthesis and signal transduction. Auxin, ethylene and abscisic acid (ABA) act as important regulators in plant growth and development. Many auxin, ethylene and ABA related genes were differentially expressed between N12 and NE2. Compared with N12, the expressions of genes encoding auxin induced protein X10A and auxin responsive protein SAUR50 were up-regulated in NE12. The expression levels of ABA signal transduction-related genes were declined, for example, protein phosphatase 2C 51 (PP2C), which is a negative regulator of the ABA response. The expressions of genes encoding ethylene-responsive transcription factor ERF023 and ERF017 were up-regulated by EBL treatment.

In addition, EBL treatment under salt stress up-regulated the expression levels of CML37 and *phyA*. CML37, calcium-binding protein, is a positive regulator of ABA during drought stress in *Arabidopsis* [25]. phyA is shown to modulate both biotic and abiotic stresses in *Arabidopsis* [26]. EBL treatment up-regulated *TIP2-1* (aquaporin TIP2-1) under salt stress (Table 2).

### 2.11. qRT-PCR Validation of Transcriptome Data

The transcription level of 33 randomly selected genes exhibiting diverse expression patterns among peanut seedlings with four treatments was investigated using qRT-PCR assay to verify the integrity of transcriptomic data. The *Ahactin* was used as the internal reference gene (Figure 10). The results confirmed that the transcription of most of the genes were consistent with the transcriptomic data of these DEGs. High correlation coefficient (R^2^ = 0.9457) was observed, suggesting the reliability of RNA-seq data (Figure 11).

## 3. Discussion

Soil salinization is one of the main abiotic stresses limiting plant growth and development [27]. As a steroidal hormone, BR not only regulates various growth and development processes, but also plays important roles in alleviating plant biotic and abiotic stress [14]. Reports on rice [28], mustard [29] and soybean [21] have shown that BR can improve plant salt tolerance. However, the regulatory mechanisms are not fully understood. We combined the results of physiology and gene expression profiling to explore the mechanism of BR in improving salt tolerance of peanut seedlings. The results of this study showed that 150 mM NaCl treatment significantly inhibited the growth of peanut seedlings, while foliar spraying of 0.1 µM EBL alleviated the salt induced inhibition through accumulation of proline and soluble sugar, and maintenance of high level of chlorophyll. At the transcriptome level, EBL up-regulated the expression of *PMP34*, *SPS2*, *BFRUCT1*, *NHX7*, *NHX8* and down-regulated the expression of *PRODH* to promote the stress adaptation.

Under salt stress, antioxidant enzymes are located in different parts of plant cells and remove excess ROS [30]. It is reported that exogenous BR can improve the salt tolerance of cucumber seedlings by increasing the activities of SOD, CAT and POD [31]. However, our results showed that there was no significant difference in SOD activity under four treatments, while the activity of CAT was significantly increased sprayed exogenous BR compared with NaCl treatment. In perennial ryegrass, exogenous 0.1 µM EBR decreased the ROS content and improved salt tolerance by increasing CAT activity [32], which was consistent with our results that 0.1 µM EBL application increased CAT activity of peanut seedlings. POD activity of peanut seedlings was elevated significantly under salt stress. When compared with salt stress treatment alone, foliar spraying of 0.1 µM EBL significantly decreased POD activity. The similar results were observed in potato. Combination of EBL and NaCl decreased the activities of SOD and POD in potato, which might be associated with the removal of the stressful conditions by the EBL treatments in the first place [33]. In the RNA-seq, we found that EBL application up-regulated genes associated with ROS scavenging including *PMP34*, *ECH2*, peroxiredoxin Q (*PrxQ*), L-ascorbate peroxidase T (*LAPX*) and glutaredoxin-C6. PMP34 is a peroxisome transporter, which is involved in the process of germination β-oxidation and auxin production [34]. ECH2 plays a role in the transformation of indole-3-butyric acid (IBA) to indole-3-acetic acid in peroxisome [35]. In chloroplast metabolism, Prx Q can not only reduce H_2_O_2_, but also catalyze the reduction of non-physiological peroxides [36]. Therefore, under NaCl stress, EBL treatment induced the activity of antioxidant enzymes leading to improve salt tolerance of peanut seedlings by up-regulating *PMP34*, *ECH2*, *Prx Q*. In addition, both L-ascorbate peroxidase T (*LAPX*) and glutaredoxin-C6 belong to ASA-GSH pathway, which were up-regulated to eliminate ROS after EBL treatment.

Proline, soluble sugar and soluble protein are important osmolytes. The concentrations of proline and soluble sugar were increased significantly after BL applied to wheat [37]. Exogenous BL application can promote the accumulation of proline and soluble sugar in apples, thus maintaining osmotic balance [38]. Consistent with these results, our results showed that compared with salt stress alone, the contents of soluble sugar and proline in peanut seedings sprayed with EBL increased significantly, but there was no difference observed in soluble protein content. RNA-seq results showed that, EBL application significantly up-regulated the expression of 14 kDa proline-rich protein DC2.15 and putative proline-rich receptor-like protein kinase PERK6, and down-regulated the expression of proline dehydrogenase 2 (*PRODH*) under salt stress. Under salt stress, spraying EBL up-regulated the expression of alpha amylase, sucrose phosphate synthase 2 and beta fructofuranosidase. The accumulated proline and soluble sugar could regulate osmotic pressure and protect plant cells.

Under salt stress, plant chl synthesis was inhibited and chl content was reduced. BRs could reduce chlorophyll degrading enzyme activity and increase chl content under abiotic stress [39]. Salt and pesticide stress led to a significant decline in chl content in rice, while the contents of chl a, chl b and total chl in salt and HBL treated seedlings were increased significantly [40]. In our study, compared with salt treatment, the contents of chl a and carotenoids were increased obviously after salt and EBL treatment. Under salt stress, application of EBL significantly increased Φ PSII, ETR and qP, while NPQ was significantly decreased, indicating that EBL could promote plant growth by promoting photosynthesis and slowing down heat dissipation. Similar results were also obtained in soybean [15] and ryegrass [18]. In addition, KEGG pathways enrichment analysis found that the most significant pathway of N12-vs.-NE12 enrichment was photosynthetic antenna protein, indicating that EBL may promote plant photosynthesis and improve salt tolerance of peanut seedlings by regulating photosynthesis related genes.

NHXs family plays the pivotal role in plant Na^+^ and K^+^ balance. Exogenous BL application reduced the accumulation of Na^+^ and increased the content of K^+^ in cells of shoots and roots under salt stress by regulating the expression of Na^+^(K^+^)/H^+^ antiporter genes [38]. EBR treatment increased K^+^, Ca^2+^, Mg^2+^ contents and decreased the ratio of Na^+^/K^+^ in perennial ryegrass under salt stress [18]. In this study, spraying EBL under salt stress up-regulated the expression level of *AhNHX7* and *AhNHX8*. These results indicated that EBL may maintain ion balance under salt stress by up regulated the expression of *NHXs*, thus improving plant salt tolerance.

EBR enhanced endogenous levels of salicylic acid (SA) and jasmonic acid (JA) while suppressing the ethylene (ETH) biosynthesis pathway in response to chilling stress [41]. However, exogenous EBR application promoted the accumulation of ABA, IAA, ZR and BR, but inhibited the levels of JA and GA_4_ under low-temperature stress in cucumber [42]. These results suggested that EBL enhanced stress tolerance through different mechanism in different plant species. In this study, the expression profile results showed that exogenous EBL application up-regulated the BR synthesis enzyme gene, BR receptor BRI1 and key factor BSK5 of BR signal pathway, thus promoting the biosynthesis and signal transduction of BR. AP2/ERF and APETALA2/ethylene response element-binding factor plays important roles in plant growth, development and environmental adaptation [43]. Under salt stress, EBL treatment up-regulated the expression of *ERF017*, *ERF023* and *PP2C51*. PP2C51 is the Key factor of abscisic acid signaling pathway. In addition, EBL treatment up-regulated the gene expression levels of auxin responsive protein SAUR50 and auxin induced protein X10A which promoted auxin signal transduction, leading to enhanced cell elongation, plant growth and development. These results suggested that under salt stress, EBL could improve the salt tolerance of peanut seedlings by regulating hormone level and signal such as BR, IAA, ABA and ethylene.

## 4. Materials and Methods

### 4.1. Plant Material and Experimental Design

#### 4.1.1. Plant Material

This experiment was conducted from April to December 2021 in greenhouse at Shandong Academy of Agricultural Sciences, Jinan, Shandong Province, China. Peanut cultivar Kainong1715 obtained from Kaifeng Agricultural and Forestry Research Institute, Henan province. Peanut seeds were washed with distilled water and soaked for 4 h and planted in plastic pot (24 cm diameter and 16 cm depth) filled with clean fine sand. Six seeds were sown in each pot, and the seedlings were watered twice a day with water until emergence. After emergence, the seedlings were watered twice a day with Hoagland solution.

#### 4.1.2. Experimental Design

Peanut seedlings of 12 d were selected and divided into four groups for treatments, CK (normal conditions), N (treated with 150 mM NaCl), NE (treated with 150 mM NaCl and 0.1 µM EBL (Sigma Chemicals, St. Louis, MI, USA)) and E (treated with 0.1 µM EBL). The NE and E treatments were treated with 0.1 µM EBL, while the CK and N treatments were treated with the same concentration of ethanol. Treatments were continued until the experiments were completed. There were five replicates in each treatment and six plants in each replicate. After 72 h of EBL pre-treatment, the salt stress was applied in N and NE treatments. After 6 d NaCl treatment, plant dry weight, fresh weight, the main stem height, root length relative water content, MDA, compatible solutes, antioxidant enzyme activities, chl content and fluorescence parameters were determined.

### 4.2. Determination of Seedling Growth and Relative Water Content (RWC) of Leaves

After 6 d treatment, the length of shoot and root were measured, five plants per treatment were measured. The aerial and root parts were separated and weighed to obtain the fresh mass. Samples were dried at 105 °C for 30 min followed by 80 °C until a constant weight was reached, to obtain the dry weight (DW).

The RWC was determined following the method described by Ghoulam et al. [44]. A fully expanded leaves were first collected and then weighed to obtain fresh weight (FW). Leaves were soaked in distilled water overnight, and weighed to obtain the turgid weight (TW). Then leaves were dried at 105 °C for 30 min followed by 80 °C, and then got the DW as described above. Calculation was done by the following formula:RWC (%) = [(FW − DW)/(TW − DW)] × 100(1)

### 4.3. Measurements of ROS and Malondialdehyde (MDA) Content

The H_2_O_2_ content was determined using an assay kit (COMINBIO, Suzhou, China) and using spectrophotometer, following the manufacturer’s instructions, and the absorbance was recorded at 415 nm. O_2_^−^ (Superoxide radicals) staining was performed using NBT staining as described by Zheng et al. [45]. Leaves of approximately 0.1 g were homogenized in liquid nitrogen for MDA measurement using thiobarbituric acid (TBA) method [46].

### 4.4. Determination of Antioxidant Enzyme Activities

Leaves of 0.5 g from peanut seedlings were used for the measurement of antioxidant enzyme activities. Superoxide dismutase (SOD) activity was determined by nitro blue tetrazolium (NBT) method [47] with minor changes. Catalase (CAT) and peroxidase (POD) activities were determined by the previously reported methods [48]. CAT activity was determined by measuring a decrease in the extinction of H_2_O_2_ at 240 nm for 2 min. POD activity was measured by guaiacol recording the change in absorbance value at 470 nm every 30 s.

### 4.5. Determination of Osmolytes Substance Content

Soluble sugar was extracting using distilled water and measured by the anthrone method [49]. Proline was extracted in sulphosalicylic acid and determined using the method of Bates et al. [50]. The total soluble protein was determined as described by Bradford et al. [51].

### 4.6. Measurements of Chl Content and Fluorescence Parameters

Leaves of 0.1 g were used for the measurements of chl content. Leaf chl was extracted in 95% ethanol, the absorbance levels were measured at 665 nm, 649 nm and 470 nm, following established methods [52]. Fluorescence parameters were measured using portable fluorometer (FMS-2 Pulse Modulated Fluorometer, Hansatech, Norfolk, UK). Leaves with the same light receiving direction and growth position were selected at 9:00 a.m. to 11:00 a.m., and minimum-adaptive fluorescence (Fo’), Steady-state fluorescence (Fs) and maximum light-adaptive fluorescence (Fm’) under light adaptation were measured. After dark adapting the leaves for 20 min, baseline (Fo) and maximum fluorescence (Fm) under dark adaptation were measured. Thereafter maximal photochemical efficiency (Fv/Fm), effective photochemical efficiency of photosystem (PS) II (Fv’/Fm’), actual photochemical efficiency of photosystem (ΦPSII), photochemical quenching (qP), nonphotochemical quenching (NPQ) were calculated [53].

### 4.7. RNA Extraction and RNA Sequencing

Total RNA of four treatments was extracted from leaves using Trizol reagent (AG RNAex Pro Reagent, Changsha, China) [54]. RNA quality and quantity were assessed by RNA agarose gel electrophoresis and spectrophotometric detection at 260 nm, respectively. The total RNA extraction was used for RNA-sequencing and real-time PCR. The RNA-sequencing was performed using DNBSEQ (BGI, Wuhan, China). The clean reads were compared to the reference sequence of cultivated peanut (GCF_003086295.2_arahy.Tifrunner.gnm1. KYV3). The differentially expressed genes (DEGs) among the selected samples were further analyzed by GO function enrichment, KEGG pathway enrichment.

### 4.8. Validation of DEGs by Quantitative Real-Time PCR (qRT-PCR)

Total RNA (1 μg) was used for reverse transcription using *Evo M-MLV* RT kit with gDNA Clean for qPCR II following the manufacturer’s instruction (AG, Changsha, China). *AhActin* was used as an internal reference gene. qRT-PCR was performed using SYBR-Green on an ABI 7500 Real-Time PCR Detection System (Applied Biosystems, Step OnePlus, Nanding, China). The relative expression level of each gene between different treatments was calculated by 2^−^^∆∆Ct^ method.

The primers used for qRT-PCR are as follows (Table 3).

### 4.9. Statistical Analysis

All experiments were arranged in a completely randomized design with three replications. Microsoft Excel 2003 is used for data processing and chart drawing. Data were analyzed by ANOVA followed by LSD method. Statistically significant differences were indicated by *p* < 0.05. Statistical computations were conducted using SPSS 23.

## 5. Conclusions

Our results showed that salt stress markedly decreased growth in peanut seedlings, while exogenous application of EBL could alleviate the inhibitory effects of salt stress. Under salt stress, exogenous application of EBL could increase the accumulation of proline and soluble sugar by down-regulating the expression of proline dehydrogenase gene and up-regulating genes related to glucose metabolism pathway. Furthermore, EBL treatment kept ion balance by up-regulating the expression of *AhNHX7* and *AhNHX8*, and scavenge reactive oxygen species by increasing peroxidase activity, accumulating glutathione, and promoting AsA-GSH cycle, resulting in the overall improvement of salt tolerance in peanut. In addition, EBL treatment could improve the RWC of peanut seedlings and thus effectively regulating osmotic stress by up-regulated the expression of aquaporin gene *TIP2-1*. These results provide insights into understanding the potential roles of BRs in peanut growth and development under salt stress.

## Figures and Tables

**Figure 1 ijms-23-06376-f001:**
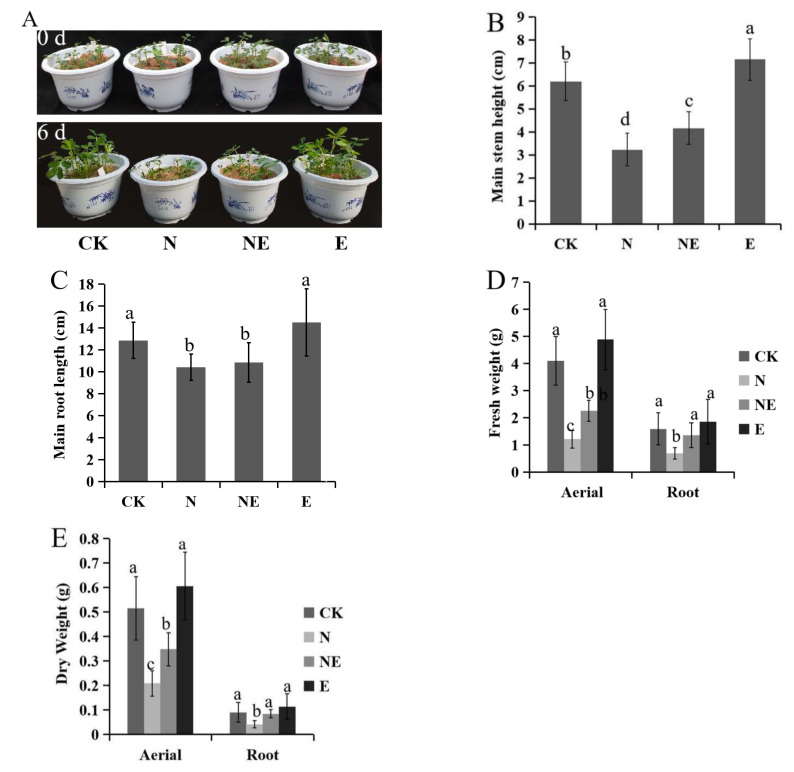
Effects of EBL on the growth of peanut seedlings under NaCl stress. CK (normal conditions), N (treated with 150 mM NaCl), NE (treated with 150 mM NaCl and 0.1 µM EBL) and E (treated with 0.1 µM EBL). (**A**) Seedling morphology; (**B**) Main stem height; (**C**) Main root length; (**D**) Seedling fresh weight; (**E**) Seedling dry weight. Different lowercase letters indicate significant differences between treatments (*p* < 0.05). The same short forms were used in the figures below.

**Figure 2 ijms-23-06376-f002:**
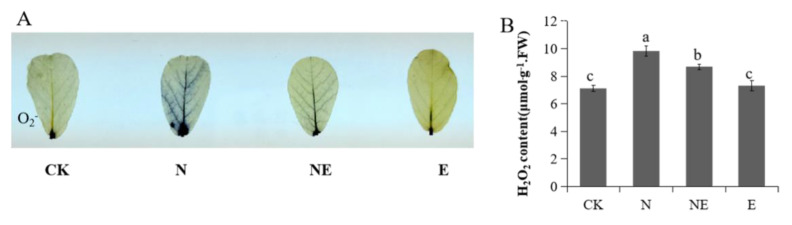
Effects of EBL on ROS production in peanut seedlings under NaCl stress. CK (normal conditions), N (treated with 150 mM NaCl), NE (treated with 150 mM NaCl and 0.1 µM EBL) and E (treated with 0.1 µM EBL). (**A**) NBT staining of superoxide anion; (**B**) H_2_O_2_ content. Different lowercase letters indicate significant differences between treatments (*p* < 0.05).

**Figure 3 ijms-23-06376-f003:**
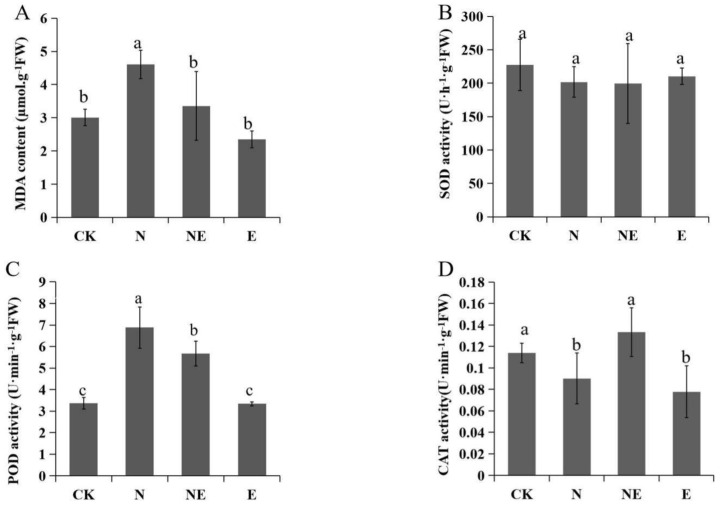
Effects of EBL on MDA content and antioxidant enzyme activities of peanut seedlings under NaCl stress. CK (normal conditions), N (treated with 150 mM NaCl), NE (treated with 150 mM NaCl and 0.1 µM EBL) and E (treated with 0.1 µM EBL). (**A**) MDA content; (**B**) SOD activity; (**C**) POD activity; (**D**) CAT activity. Different lowercase letters indicate significant differences between treatments (*p* < 0.05).

**Figure 4 ijms-23-06376-f004:**
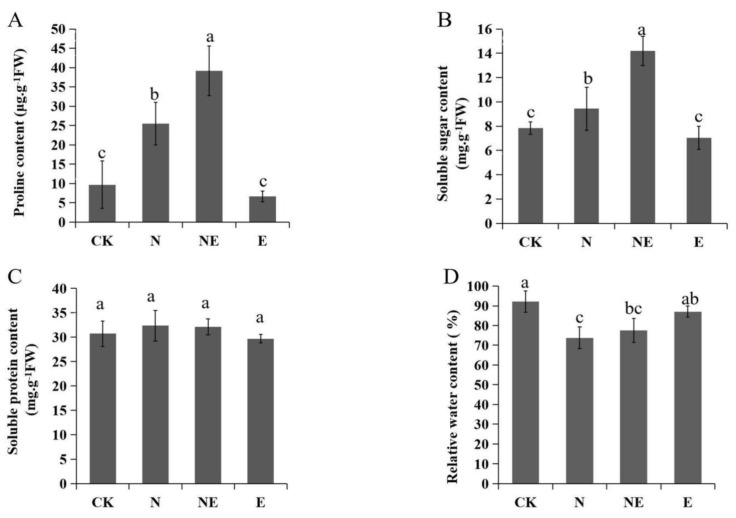
Effects of EBL treatments on osmolytes content and RWC of peanut seedling under salt stress. CK (normal conditions), N (treated with 150 mM NaCl), NE (treated with 150 mM NaCl and 0.1 µM EBL) and E (treated with 0.1 µM EBL). (**A**) Soluble sugar content; (**B**) Proline content; (**C**) Soluble protein content; (**D**)Relative water content. Different lowercase letters indicate significant differences between treatments (*p* < 0.05).

**Figure 5 ijms-23-06376-f005:**
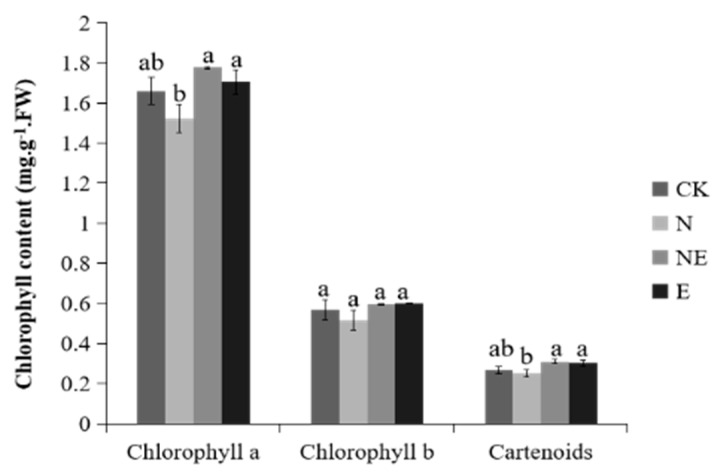
Effects of EBL on chlorophyll and carotenoids content of peanut seedling under salt stress. CK (normal conditions), N (treated with 150 mM NaCl), NE (treated with 150 mM NaCl and 0.1 µM EBL) and E (treated with 0.1 µM EBL). Different lowercase letters indicate significant differences between treatments (*p* < 0.05).

**Figure 6 ijms-23-06376-f006:**
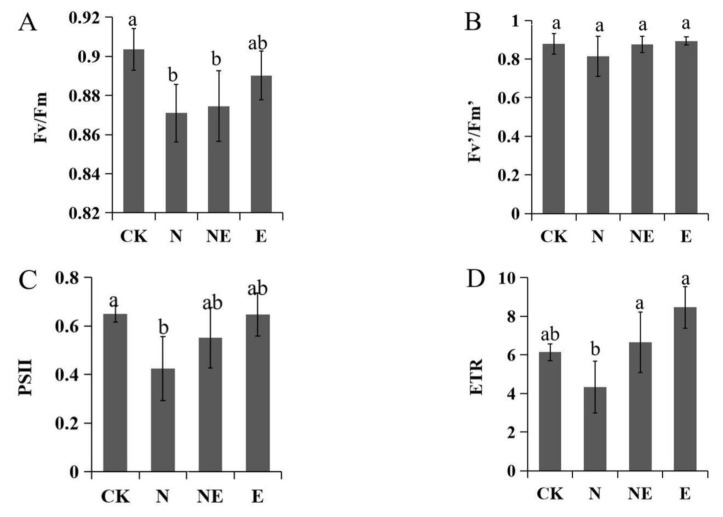
Effects of EBL treatments on chlorophyll fluorescence parameters of peanut seedling under salt stress. CK (normal conditions), N (treated with 150 mM NaCl), NE (treated with 150 mM NaCl and 0.1 µM EBL) and E (treated with 0.1 µM EBL). (**A**) Fv/Fm; (**B**) Fv’/Fm’; (**C**) PSII; (**D**) ETR; (**E**) qP; (**F**) NPQ. Different lowercase letters indicate significant differences between treatments (*p* < 0.05).

**Figure 7 ijms-23-06376-f007:**
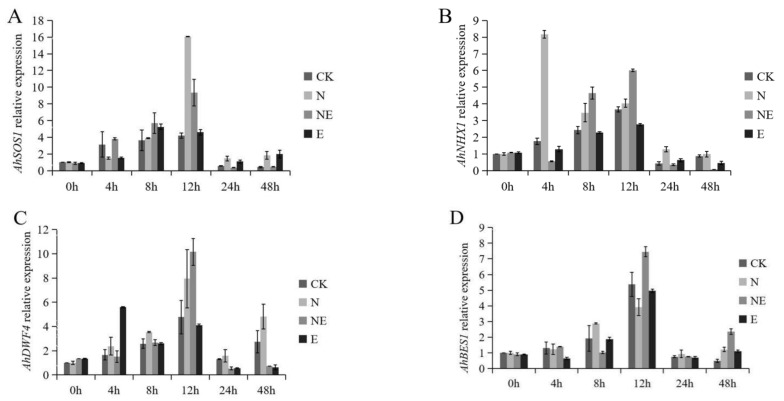
Effects of EBL treatments on key genes relative expression of peanut seedling under salt stress. CK (normal conditions), N (treated with 150 mM NaCl), NE (treated with 150 mM NaCl and 0.1 µM EBL) and E (treated with 0.1 µM EBL). (**A**) *AhSOS1* relative expression; (**B**) *AhNHX1* relative expression; (**C**) *AhDWF4* relative expression; (**D**) *AhBES1* relative expression.

**Figure 8 ijms-23-06376-f008:**
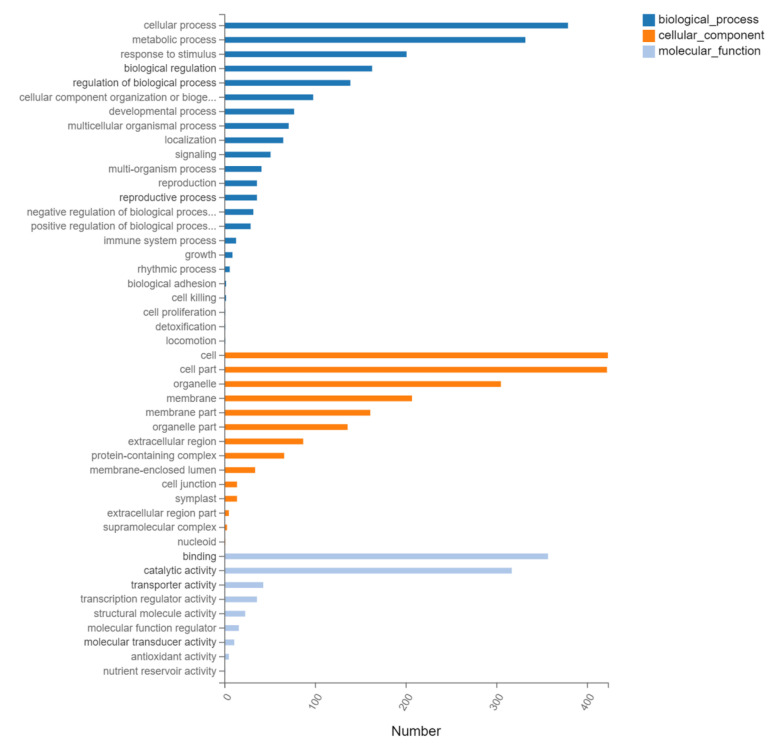
GO classification of differentially expressed genes.

**Figure 9 ijms-23-06376-f009:**
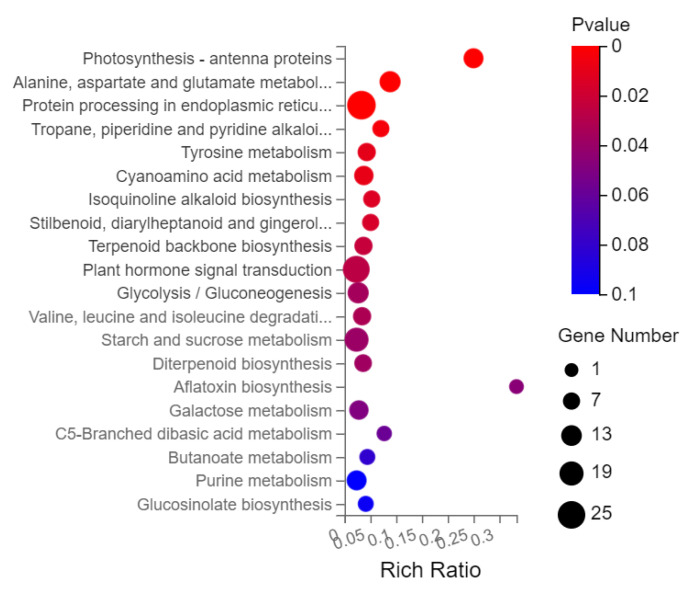
Bubble diagram of KEGG pathway enrichment of differentially expressed genes.

**Figure 10 ijms-23-06376-f010:**
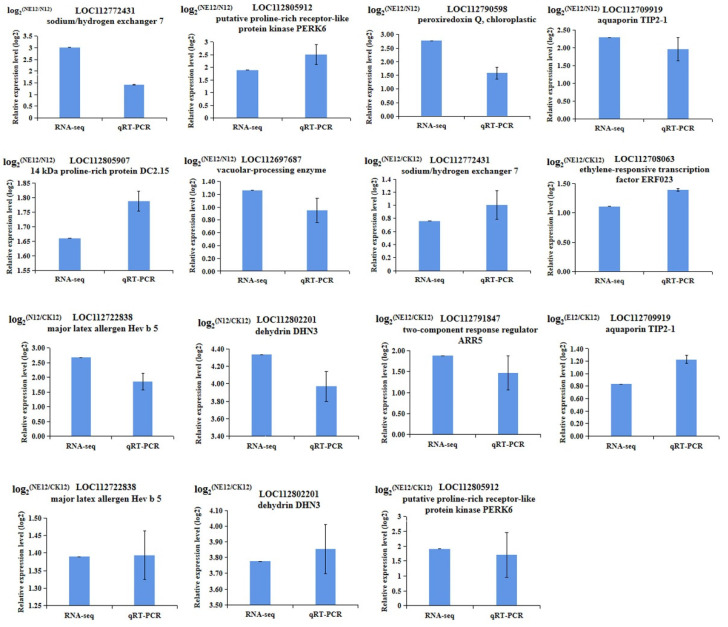
Verification of differentially expressed genes by qRT-PCR.

**Figure 11 ijms-23-06376-f011:**
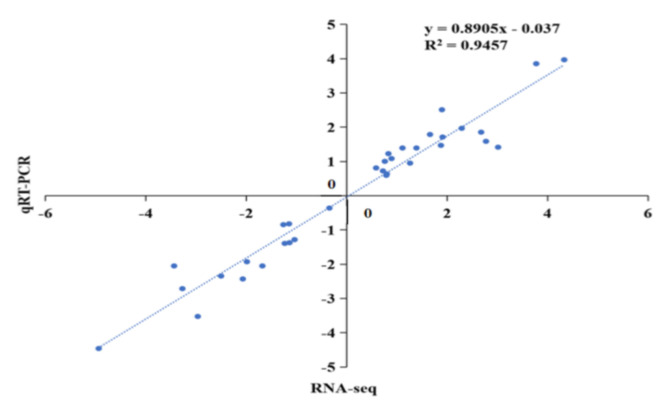
Pearson’s correlation of RNA-seq and qRT-PCR results.

**Table 1 ijms-23-06376-t001:** RNA-Seq quality statistics.

Sample	Total Raw Reads (M)	Total Clean Reads (M)	Total Clean Bases (Gb)	Clean Reads Q20 (%)	Clean Reads Q30 (%)
CK12_1	23.92	23.8	1.19	98.42	95.18
CK12_2	23.92	23.83	1.19	98.28	94.72
CK12_3	23.92	23.8	1.19	98.3	94.8
N12_1	23.92	23.78	1.19	98.38	95.03
N12_2	23.92	23.6	1.18	98.45	95.26
N12_3	23.92	23.76	1.19	98.3	94.86
NE12_1	23.92	23.81	1.19	98.38	95.06
NE12_2	23.92	23.78	1.19	98.39	95.12
NE12_3	23.92	23.8	1.19	98.39	95.06
E12_1	23.92	23.81	1.19	98.43	95.2
E12_2	23.92	23.78	1.19	98.38	95.05
E12_3	23.92	23.78	1.19	98.39	95.09

**Table 2 ijms-23-06376-t002:** Expressed changes of salt tolerance genes.

Gene ID	log_2_ ^FC^ (NE12/N12)	Q Value	Symbol	Gene Bank Description
Na^+^/H^+^ antiporters (NHXs)				
112772431	3.02	1.39 × 10^−10^	NHX7	sodium/hydrogen exchanger 7
112732776	5.59	4.18 × 10^−19^	NHX8	sodium/hydrogen exchanger 8
ROS scavenging related genes				
112696745	2.52	7.43 × 10^−6^	PMP34	peroxisomal nicotinamide adenine dinucleotide carrier
112733788	5.91	5.07 × 10^−^^3^	AAE11	butyrate-CoA ligase AAE11, peroxisomal
112766271	9.13	2.49 × 10^−^^11^	ECH2	enoyl-CoA hydratase 2, peroxisomal
112790598	2.78	1.3 × 10^−^^26^	PRXQ	peroxiredoxin Q, chloroplastic
112701439	1.58	7.05 × 10^−^^3^	LAPX	L-ascorbate peroxidase T, chloroplastic
112706364	1.76	1.49 × 10^−^^2^		glutaredoxin-C6
osmotic stress related genes				
112797943	1.10	1.31 × 10^−^^4^	PERK9	proline-rich receptor-like protein kinase PERK9
112805912	1.90	3.86 × 10^−^^5^	PERK6	putative proline-rich receptor-like protein kinase PERK6
112744691	−1.44	2.56 × 10^−^^2^	PRODH	proline dehydrogenase 2, mitochondrial
112797983	−1.36	1.86 × 10^−^^5^	PRODH	proline dehydrogenase 2, mitochondrial
112697841	−1.69	5.96 × 10^−^^4^	AMY	alpha-amylase-like
112697429	1.04	6.37 × 10^−^^5^	SWEET1	bidirectional sugar transporter SWEET1
112724011	5.81	7.7 × 10^−^^3^	SUC	sugar carrier protein C-like
112741002	1.13	7.17 × 10^−^^7^	SPS	probable sucrose-phosphate synthase 2
112776149	3.77	4.28 × 10^−^^44^	BFUCT	beta-fructofuranosidase, cell wall isozyme
phytohormones related genes				
112715407	1.94	1.17 × 10^−^^2^	CPD	cytochrome P450 90A1
112709363	6.72	1.44 × 10^−^^4^	BRI1	squamosa promoter-binding-like protein 6
112792599	−1.03	3.39 × 10^−^^2^	BRI1	squamosa promoter-binding-like protein 12
112788794	1.43	1.04 × 10^−^^2^	BSK5	serine/threonine-protein kinase BSK5
112697317	−2.30	4.5 × 10^−^^10^	TCH4	probable xyloglucan endotransglucosylase/hydrolase protein 23
112698421	1.85	1.86 × 10^−^^5^	TCH4	xyloglucan endotransglucosylase/hydrolase protein 22
112727310	1.41	8.28 × 10^−^^4^	TCH4	probable xyloglucan endotransglucosylase/hydrolase protein 23
112736986	2.18	3.12 × 10^−^^13^	X10A	auxin-induced protein X10A
112767282	2.84	4 × 10^−^^2^	SAUR50	auxin-responsive protein SAUR50
112709815	1.38	6.63 × 10^−^^7^		gibberellin 3-beta-dioxygenase 1
112755883	−1.18	6.96 × 10^−^^5^		cytokinin dehydrogenase 7
112749719	−1.26	4 × 10^−^^3^	PP2C51	protein phosphatase 2C 51
112708063	4.34	1 × 10^−^^2^	ERF023	ethylene-responsive transcription factor ERF023
112738367	4.95	1 × 10^−^^2^	ERF017	ethylene-responsive transcription factor ERF017
the others				
112708162	2.00	1.41 × 10^−^^2^	CML37	calcium-binding protein CML37
112710657	3.36	1.21 × 10^−^^29^	PhyA	phytochrome A
112709919	2.29	2.49 × 10^−5^	TIP2-1	aquaporin TIP2-1

**Table 3 ijms-23-06376-t003:** The following primers were used in this experiment.

Gene ID	Primer Name	Primer Sequence (5′–3′)
112772431	1-NHX7-1F	GTTCGCTTTACACTACCTTGAC
	1-NHX7-1R	TTCTCTAAGTTCCTCATCATCTCC
112805907	3-DC2.15-3F	AATTCTTGTTGTGTCACTCC
	3-DC2.15-3R	CAAGTCCTAAGACATCAGCA
112805912	4-PERK6-4F	ATTCTCAAACTGGTTCATCCAG
	4-PERK6-4R	AAACAGGTAATCCAAAGCCA
112790598	9-PERQ-9F	TTTCTATCCTGCTGATGAGTCC
	9-PERQ-9R	TCACTACTGATTCCAACAACCT
112709919	6-TIP2-1-6F	ACTTTCTGGAATCTTCTACTGG
	6-TIP2-1-6R	TCACTACTCCTTCAAATGCTC
112708063	15-ERF023-15F	CATGGAGCTACAAACAACGG
	15-ERF023-15R	AATCTCCGAAACCCATTTCC
112779663	1-GPS3-1F	GAAGCCAATCGTACATTTCC
	1-GPS3-1R	AAGTTGTGGATAAGAGAAGTGG
112801730	2-CP26-2F	ATTCCACACCCTTACTACTTCC
	2-CP26-2R	GTTTCCAAGCATCTCAGACAC
112736176	3-CP4-3F	GAAAGGACGAACAAGAACCA
	3-CP4-3R	GTTGTTACAGTCGCCATCTC
112767125	4-CP215-4F	CTTTCTACTCTCAACCTTCACTC
	4-CP215-4R	TGCGGAGAAGTTCATTTCCT
112697687	5-VPE-5F	GTCCCTTCTAAGGATCACCC
	5-VPE-5R	TGGCTCATGGAAGAATCTGG
112707104	6-R7OM-6F	TCGTGCATTACCTGAATACC
	6-R7OM-6R	GAACCTTGTGAACCATTGAG
112792338	7-LHC-7F	CACTCACTCATCAACACCAC
	7-LHC-7R	TCTTCCAACTCCAAGCTCAG
112703831	8-LEA5-8F	GGTAAGAGGAAGTTGGTTTATGTG
	8-LEA5-8R	CTGGATTCTAATGTTGCCTGTG
112802201	9-DHN3-9F	AGTATGGCAACACTATGAGG
	9-DHN3-9R	GTACATCTTGCCGGATTCAG
112696380	10-GIP1-10F	CCTGCCTCATCAACTATAAATACC
	10-GIP1-10R	GAGAAGAGAAGCAAGAAGGG
112722388	11-SNA2-11F	GTAGAAGCGAGGAATCACAG
	11-SNA2-11R	AAGGCAGAGAAGCAATAACAC
112730690	12-EN75-12F	CATTGCTGATGAATACCCTAAACC
	12-EN75-12R	TATGGTGGCTTCTCATGTGG
112722838	13-MLA5-13F	CCCAACCCATTCAATTCTTCC
	13-MLA5-13R	TTCTGTGGTTGTTTCCTCTG
112791847	14-ARR5-14F	ATTCCCTCTCAATCTAACGG
	14-ARR5-14R	AGGACATAATCACAACTGGA
AhActin	ActinF	GTCATCGTCATCCTCTTCTC
	ActinR	CATTCCTGTTCCATTGTCAC
AhSOS1	1-SOS-F	GAGATTTCCCTTACACTTGCC
	1-SOS-R	GAACATTCCCAACGACATGAC
AhNHX1	3-NHX1-F	CCGCCTATAATATTCAATGCCG
	3-NHX1-R	CCGAAGGTTATGATGGTACAC
AhDWF4	4-DWF4-F	TTATTCTGCTACCACCAT
	4-DWF4-R	CATCTGCTGACACTATTG
arahy.7MK8W0.1	9-BES1-F	AATCCCTCTCCAATGTCTCC
	9-BES1-R	GGAAGCATCAGATTCATCACAC

## Data Availability

The data associated with this manuscript can be found in the Sequence Read Archive (SRA) at NCBI under the BioProject submission ID: SUB11566623 after the release.

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
