# Peer review of "The Mechanisms Underlying Salt Resistance Mediated by Exogenous Application of 24-Epibrassinolide in Peanut"

_ijms, 2022, doi:10.3390/ijms23126376_

Round 1

Reviewer 1 Report

The manuscript "The mechanisms underlying salt resistance..." by Li et al. is an interesting study regarding the effect of epibrassinolide on the stress responses of peanut. The paper provides not only basic growth analysis, but adds a thorough and well-planned transcriptome analysis.

I have a number of hopefully minor comments:

  • a thorough language check should be useful 
    • line 27 900 million hectares of land WERE affected,
    • line 28 ... concentration in THE plant growth env....,
    • line 36 A high level...,
    • line 37-38 and 41-42 Salt stress LEAD
    • line 45 triggered A calcium signal ... THE SOS2SOS3 protein kinase
    • line 229 "...a NEGATIVE regulator OF the ABA..."
    • ...
  • The authors describe biomass reduction but did not note (or at least mention) changes in plant architecture (more lateral roots, more shoot branching), as presumed for low stress levels (see Stress Induced Plant Morphogenesis, SIMR, eg. in Patakas, 2012 or Jansen et al. 2012). 
  • Figure 1D: legend is missing, I assume it is the same as in 1E ?
  • How were significant changes of ROS determined on images as in Fig 2A.
  • Given the error bars in fig 6, rounding off at two digits behind the digital point in , e.g. line 150 seems excessively precise.
  • Table 2: how were these genes ordered in the table ?
  • is there a relation between the gene expression data in figure 10 and the data in figure 11 ?I cannot find, say, any graph in Fig 11 with a RNA-seq of around -4.5 and a qRT-PCR of around -5 (the
  • Figure 11 is also not referenced in the text as far as I can find.
  • Authors should provide data on repetitions, plant numbers used....

Patakas, A. (2012). Abiotic stress-induced morphological and anatomical changes in plants. In Abiotic stress responses in plants (pp. 21-39). Springer, New York, NY.

Jansen, M. A., Coffey, A. M., & Prinsen, E. (2012). UV-B induced morphogenesis: four players or a quartet?. Plant Signaling & Behavior, 7(9), 1185-1187.

Author Response

Dear Editor and Reviewers,

Thank you very much for the opportunity to revise our manuscript (ijms-1703179) entitled "The mechanisms underlying salt resistance mediated by exogenous application of 24-epibrassinolide in peanut". We value the comments received greatly, as they have pointed out important issues before further processing. We have now incorporated several modifications to the initial manuscript following your comments and suggestions. The modified and added texts in the revised manuscript were marked up using the “Track Changes” function. Following this letter, the reviewer’s comments are reproduced in black with our point-by-point responses in blue. 

The manuscript "The mechanisms underlying salt resistance..." by Li et al. is an interesting study regarding the effect of epibrassinolide on the stress responses of peanut. The paper provides not only basic growth analysis, but adds a thorough and well-planned transcriptome analysis.

(1) a thorough language check should be useful 

  • line 27 900 million hectares of land WERE affected,
  • line 28 ... concentration in THE plant growth env....,
  • line 36 A high level...,
  • line 37-38 and 41-42 Salt stress LEAD
  • line 45 triggered A calcium signal ... THE SOS2SOS3 protein kinase
  • line 229 "...a NEGATIVE regulator OF the ABA..."
  • ...

Answer:Thank you for your suggestions. We revised it in the revised version.

  • The authors describe biomass reduction but did not note (or at least mention) changes in plant architecture (more lateral roots, more shoot branching), as presumed for low stress levels (see Stress Induced Plant Morphogenesis, SIMR, eg. in Patakas, 2012 or Jansen et al. 2012). 

Patakas, A. (2012). Abiotic stress-induced morphological and anatomical changes in plants. In Abiotic stress responses in plants (pp. 21-39). Springer, New York, NY.

Jansen, M. A., Coffey, A. M., & Prinsen, E. (2012). UV-B induced morphogenesis: four players or a quartet?. Plant Signaling & Behavior, 7(9), 1185-1187.

Answer:Thank you for your suggestions!After 6 d treatment, the number of lateral roots and shoot branching were not changed significantly, but plant dry weight, fresh weight, the main stem height and root length were changed significantly. Therefore, we replaced biomass with dry weight, fresh weight, main stem height, and root length.

(3) Figure 1D: legend is missing, I assume it is the same as in 1E ?

Answer:Thank you for your suggestions. Yes, as you assumed, the legend of Figure 1D is the same as Figure 1E. And the legend of Figure 1D was added.

  • How were significantchanges of ROS determined on images as in Fig 2A.

Answer:Thank you for your suggestions! NBT can react with superoxide anion (O2·-) to form blue formazan. The intensity of the blue color indicates the amount of superoxide anions. As shown in Fig. 2A, the blue color intensity in leaves treated with 150 mM NaCl was much higher than those of control plants, indicating more O2·− in NaCl treated plants than that in control plants. No quantification of the blue color was carried out and therefore, we remove “significant” in the revised MS.

  • Given the error bars in fig 6, rounding off at two digits behind the digital point in , e.g. line 150 seems excessively precise.

Answer:Thank you for your suggestions!In the text, the part was revised as “Under 150 mM NaCl treatment, the maximum photochemical quantum yield of PSII (Fv/Fm), PSII (ΦPSII), electron transfer efficiency (ETR), and photochemical quenching coefficient (qP) were decreased significantly to compare with the control. While salt treatment led to an increase of non-photochemical quenching (NPQ) remarkably. EBL treatment increased the values of ΦPSII, ETR and qP, respectively, comparing with only NaCl treatment. However, EBL application on plants treated by NaCl led to a decrease of NPQ”.

  • Table 2: how were these genes ordered in the table ?

Answer:Thank you for your suggestions!These genes in the table were grouped in Genes encoding Na+/H+ antiporters (NHXs), ROS scavenging related genes, osmotic stress related genes, phytohormones related genes and the others. We added the illustration in the revised table.

  • Is there a relation between the gene expression data in figure 10 and the data in figure 11? I cannot find, say, any graph in Fig 11 with a RNA-seq of around -4.5 and a qRT-PCR of around -5 (the Figure 11 is also not referenced in the text as far as I can find.

Answer:Thank you for your suggestions. Figure 10 (Original figure 11) added in line 267. The gene expression data in  Figure 11 (Original figure 10) were corresponding to the data in Figure 10 (Original figure 11). The data of A RNA-seq of around -4.5 and a qRT-PCR of around -5 was the expression data of the gene encoding 14 kDa proline-rich protein DC2.15 by calculating log2 (N12 / CK12)

  • Authors should provide data on repetitions, plant numbers used....

Answer:Thank you for your suggestions. We revised it in the revision.

Thank you and best regards.

Yours sincerely,
Shuzhen Zhao

Institute of Crop Germplasm Resources

Shandong Academy of Agricultural Sciences,

Ji’nan 250100, P. R. China

Email: zhaoshuzhen51@126,com

Reviewer 2 Report

The study lacks merit for publication due to following reasons:

(1) Soil salinity is one of the major abiotic stress limiting ground nut crop productivity. However, the authors have used only one concentration of stress i.e., 150 mM NaCl which is not sufficient to arrive at a conclusions the study projected.

(2) Secondly, the authors should have tried multiple concentrations of EBL to understand the mechanisms the study aimed for. 

(3) The study also aimed to identify the differential expression of genes under salinity stress and have used only one genotype which is absolutely insufficient as genotypic differences are not accounted in arriving at the conclusion. 

(4) Above all, the studied parameters were quantified only at one stage of crop growth i,e., 6 days after NaCl treatment which is again a narrow way of looking at the effect of salinity stress.

With only one concentration of EBL spray at only one condition of salinity stress at a single stage of crop growth of only one genotype of groundnut will not yield a meaningful conclusion. The materials used are insufficient and the methodology followed is inadequate to arrive at a conclusion the authors presented in the manuscript. Hence the manuscript is REJECTED.

Author Response

Dear Editor and Reviewers,

Thank you very much for the opportunity to revise our manuscript (ijms-1703179) entitled "The mechanisms underlying salt resistance mediated by exogenous application of 24-epibrassinolide in peanut". We value the comments received greatly, as they have pointed out important issues before further processing. We have now incorporated several modifications to the initial manuscript following your comments and suggestions. The modified and added texts in the revised manuscript were marked up using the “Track Changes” function. Following this letter, the reviewer’s comments are reproduced in black with our point-by-point responses in blue. 

  • Soil salinity is one of the major abiotic stress limiting ground nut crop productivity. However, the authors have used only one concentration of stress i.e., 150 mM NaCl which is not sufficient to arrive at a conclusions the study projected.

Answer:Thank you for your suggestions. In fact, we treated peanut seedlings of 12 d with five NaCl concentrations, 0 mM, 50 mM, 100 mM, 150 mM and 200 mM. According to the results including plant height and MDA content under different NaCl concentrations, 150 mM NaCl was selected as the stress concentration. We added these results in the revised version.

  • Secondly, the authors should have tried multiple concentrations of EBL to understand the mechanisms the study aimed for. 

Answer:Thank you for your suggestions. On the basis of 150 mM NaCl, we also carried out five EBL concentration treatments. We added the results in the revised version.

  • The study also aimed to identify the differential expression of genes under salinity stress and have used only one genotype which is absolutely insufficient as genotypic differences are not accounted in arriving at the conclusion. 

Answer: Thank you for your suggestions. There may be different in salt response among different genotypes of peanut. In this study, the purpose is to illuminate the mechanisms underlying salt resistance mediated by exogenous application of 24-epibrassinolide in peanut, not to compare the difference in salt and 24-epibrassinolide responses of different varieties. Furthermore, we referred to many relevant literatures, which also used one variety to carry out similar studies. Some literatures are as followed:

  • Su QF,Zheng XD, Tian YK et al. Exogenous brassinolide alleviates salt stress in Malus hupehensis by regulating the transcription of NHX-Type Na+(K+)/H+ antiporters. Front. Plant Sci. 2020, 11, 38. 11:38.
  • Li J, Yang P, Kang JG et al. Calderón-Urrea, A.; Lyu, J.; Zhang, G.B.; Feng, Z.; Xie, J.J. Transcriptome analysis of pepper (Capsicum annuum) revealed a role of 24-Epibrassinolide in response to chilling. Plant Sci.2016, 7, 1281.
  • Soliman M, Elkelish A, Souad T et al. Brassinosteroid seed priming with nitrogen supplementation improves salt tolerance in soybean. Physiol Mol Biol Plants, 2020, 26: 501-511.
  • Huang L, Zhang L, Zeng R et al. Brassinosteroid Priming Improves Peanut Drought Tolerance via Eliminating Inhibition on Genes in Photosynthesis and Hormone Signaling. Genes (Basel), 2020, 11: undefined.
  • Dou L, Sun Y, Li S et al. Transcriptomic analyses show that 24-epibrassinolide (EBR) promotes cold tolerance in cotton seedlings. PLoS One, 2021, 16: e0245070.
  • Anket S, Sharad T, Vinod K et al. 24-epibrassinolide stimulates imidacloprid detoxification by modulating the gene expression of Brassica juncea L. BMC Plant Biol, 2017, 17: 56.
  • Above all,the studied parameters were quantified only at one stage of crop growth i,e., 6 days after NaCl treatment which is again a narrow way of looking at the effect of salinity stress.

Answer: In our study, we observed the grow inhibition after 0, 3, and 6 day of treatments using NaCl and EBL. NaCl inhibition and EBL alleviation of the inhibition was not obvious at day 3, but obvious at day 6. Therefore, the studied parameters were quantified only at day 6 after treatments. The following two papers using the similar strategy as ours. They used one point of treatment for the detailed analysis.

  • Su QF, Zheng XD, Tian YK, et al. Exogenous brassinolide alleviates salt stress in Malus hupehensis by regulating the transcription of NHX-Type Na+(K+)/H+antiporters. Front. Plant Sci. 2020, 11, 38. 11:38.
  • Li J, Yang P, Kang JG, Gan YT et al. Transcriptome analysis of pepper (Capsicum annuum) revealed a role of 24-Epibrassinolide in response to chilling. Plant Sci.2016, 7, 1281.

Thank you and best regards.

Yours sincerely,
Shuzhen Zhao

Institute of Crop Germplasm Resources

Shandong Academy of Agricultural Sciences,

Ji’nan 250100, P. R. China

Email: zhaoshuzhen51@126,com
